# Redistribution and Activation of CD16brightCD56dim NK Cell Subset to Fight against Omicron Subvariant BA.2 after COVID-19 Vaccination

**DOI:** 10.3390/microorganisms11040940

**Published:** 2023-04-03

**Authors:** Huiyun Peng, Tianxin Xiang, Fei Xu, Yuhuan Jiang, Lipeng Zhong, Yanqi Peng, Aiping Le, Wei Zhang, Yang Liu

**Affiliations:** 1Departments of Clinical Laboratory, Medical Center of Burn Plastic and Wound Repair, The First Affiliated Hospital of Nanchang University, Nanchang 330006, China; huiyun.peng@ncu.edu.cn (H.P.);; 2National Regional Center for Respiratory Medicine, China-Japan Friendship Hospital Jiangxi Hospital, Nanchang 330200, China; 3Department of Hospital Infection Control, The First Affiliated Hospital of Nanchang University, Nanchang 330006, China; 4Department of Respiratory Medicine, The First Affiliated Hospital of Nanchang University, Nanchang 330006, China; 5Department of Transfusion, The First Affiliated Hospital of Nanchang University, Nanchang 330006, China

**Keywords:** Omicron subvariant BA.2, COVID-19, NK cell subset, vaccination, immune responses

## Abstract

With the alarming surge in COVID-19 cases globally, vaccination must be prioritised to achieve herd immunity. Immune dysfunction is detected in the majority of patients with COVID-19; however, it remains unclear whether the immune responses elicited by COVID-19 vaccination function against the Omicron subvariant BA.2. Of the 508 enrolled patients infected with Omicron BA.2, 102 were unvaccinated controls, and 406 were vaccinated. Despite the presence of clinical symptoms in both groups, vaccination led to a significant decline in nausea or vomiting, abdominal pain, headache, pulmonary infection, and overall clinical symptoms and a moderate rise in body temperature. The individuals infected with Omicron BA.2 were also characterised by a mild increase in both serum pro- and anti-inflammatory cytokine levels after vaccination. There were no significant differences or trend changes between T- and B-lymphocyte subsets; however, a significant expansion of NK lymphocytes in COVID-19-vaccinated patients was observed. Moreover, the most effective CD16^bright^CD56^dim^ subsets of NK cells showed increased functional capacities, as evidenced by a significantly greater IFN-γ secretion and a stronger cytotoxic potential in the patients infected with Omicron BA.2 after vaccination. Collectively, these results suggest that COVID-19 vaccination interventions promote the redistribution and activation of CD16^bright^CD56^dim^ NK cell subsets against viral infections and that they could facilitate the clinical management of patients infected with Omicron BA.2.

## 1. Introduction

The COVID-19 outbreak continues to spread widely, posing a great threat to physical and mental health and presenting a massive economic burden. Globally, as of 6:27 pm CET, 8 February 2023, there have been 755,041,562 confirmed cases of COVID-19, including 6,830,867 deaths, reported to the World Health Organization (WHO) (WHO COVID-19 Dashboard, 2023). Natural selection favouring more infectious variants was discovered as the fundamental law of biology governing SARS-CoV-2 transmission and evolution, including the occurrence of the Alpha (B.1.1.7), Beta (B.1.351), Gamma (P.1), Delta (B.1.617.2), Kappa (B.1.617.1), Epsilon (B.1.427/B.1.429), Lambda (C.37), and Omicron variants (B.1.1.529) [1,2]. The emergence of the SARS-CoV-2 Omicron variant of concern (VOC) in November 2021 coincided with the fourth major outbreak of the COVID-19 pandemic [3,4]. Omicron variants have been classified into four subspecies: BA.1, BA.1.1, BA.2, and BA.3. The spread of BA.2, first described in South Africa, differed greatly by geographic region, in contrast to BA.1, which followed a similar global expansion, firstly occurring in Asia and subsequently in Africa, Europe, Oceania, and North and South America [5]. Omicron BA.2 is also substantially more transmissible than BA.1 and is capable of vaccine resistance [6]. A comparative analysis of all the main variants revealed that BA.2 is approximately 4.2- and 1.5-fold more contagious than Delta and BA.1, respectively, and approximately 17 times and 30% more capable than Delta and BA.1, respectively, of evading vaccine protection [7]. Although Omicron BA.2 did not result in more severe disease than BA.1, its reinfection is astonishing. This means that the antibodies generated from the early Omicron BA.1 strain infection were evaded by the BA.2 subvariant.

BA.2 shares 32 mutations with BA.1 but has 28 distinct mutations, while it has 4 unique mutations and 12 mutations shared with BA.1 on the receptor-binding domain (RBD) [7]. Omicron BA.1 is widely known for its ability to escape current vaccines [8,9]. To date, no observations regarding the infectivity, vaccine breakthrough, or antibody resistance of the BA.2 strain have been reported [6]. The emergence of Omicron BA.2 has posed new requirements and challenges in combating SARS-CoV-2. A large number of mutations in the spike protein indicate that its response to immune protection triggered by the existing SARS-CoV-2 infection and vaccines may be altered. Whether the Omicron variant BA.2 results in more infectious or serious symptoms than other variants remains poorly understood, and whether it can evade immune responses elicited by vaccination or natural infection has become the greatest concern.

To prevent COVID-19, several types of COVID-19 vaccines are being developed in China, including inactivated vaccines, viral vector vaccines, and protein subunit vaccines [10]. Of these, China’s CoronaVac and BBIBP-CorV inactivated vaccines have been widely used in more than 110 countries and account for nearly half of the COVID-19 vaccine doses administered globally, and they show good safety and immunogenicity after immunisation in humans [11,12,13], further highlighting the important role of inactivated vaccines in the control of the pandemic. Different people may have different antibodies and immune responses to the same vaccine, depending on gender, age, race, and underlying medical conditions. Previous studies have shown that abnormalities in lymphocyte subsets and cytokine storms facilitate the onset and progression of COVID-19 infection [14,15,16]. However, it remains unclear whether the immune responses elicited by COVID-19 vaccination act against the Omicron subvariant BA.2. Therefore, understanding the vaccine escape potential of Omicron BA.2 is important.

The primary objectives of this study were to describe the variations in peripheral blood lymphocyte subsets and cytokine profiles in patients infected with Omicron subvariant BA.2 after vaccination and to investigate the effects of these immune parameters on the hospital admission and clinical symptoms of individuals infected with Omicron BA.2.

## 2. Materials and Methods

### 2.1. Participants and Study Design

A total of 537 patients, not previously exposed to other SARS-CoV-2 variants of concern, were confirmed to have Omicron subvariant BA.2 infection via viral nucleic acid testing (Da An Gene Co., Ltd., Guangzhou, China and Bio-Germ Co., Ltd., Shanghai, China) at the First Affiliated Hospital of Nanchang University for isolation and treatment between 20 March 2022 and 30 April 2022. All cases were sent to the Jiangxi provincial CDC for further sequencing using an Illumina Nextseq 550 sequencing platform. Of these, 29 individuals were excluded because of the absence of lymphocyte subpopulations and cytokine profile data; consequently, 508 patients were included in the final analysis. All patients infected with Omicron BA.2 were divided into an unvaccinated group and a vaccinated group, with patients who received 1–3 doses of COVID-19 inactivated vaccines (CoronaVac developed by Sinovac and BBIBP-CorV developed by Sinopharm) identified according to a vaccination history record. Among all patients, 102 (49 males and 53 females) were unvaccinated controls, with a median age of 33 years (3.5~70.5), and 406 (190 males and 216 females) were vaccinated, with a median age of 47 years (19.0~66.0). All patients had symptomatic infections without other serious illnesses. The patients’ peripheral blood samples were drawn on the first day of admission to analyse the immune response. Each participant provided signed informed consent before participating in the study. This study was approved by the Ethics Committee of the First Affiliated Hospital of Nanchang University and was performed in compliance with the Declaration of Helsinki (Approval Code: (2023) CDYFYYLK (01-019); Approval Date: 29 January 2023).

### 2.2. Lymphocyte Subset Detection

Peripheral venous blood samples (EDTA anticoagulated) were collected from all participants. The absolute numbers and percentages of CD3+ T cells, CD4+ T cells, CD8+ T cells, B cells, and NK cells were determined using a 6-color TBNK Reagent Kit (anti-CD45-PerCP-Cy5.5 (2D1), anti-CD3-FITC (UCHT1), anti-CD4-PC7 (RPA-T4), anti-CD8-APC-Cy7 (H1T8a), anti-CD19-APC (H1B19), anti-CD16-PE (CB16), and anti-CD56-PE (MEM-188)) (QuantoBio Technology, Beijing, China) with QB cell-count tubes, according to the manufacturer’s instructions. The kit uses a lyse-no-wash staining procedure and provides absolute cell numbers. At 37 °C, 50 μL of whole blood was stained with 20 μL of a 6-color TBNK antibody cocktail for 15 min. After adding 450 μL of RBC lysis solution (QuantoBio Technology, Beijing, China) and after 15 min of incubation, the samples were analysed using a DxFLEX flow cytometer (Beckman Coulter, Fullerton, CA, USA). All data were analysed using FlowJo software (version 10.5.3, FlowJo LLC, Ashland, OR, USA).

NK cell subsets. The following antibodies were purchased from Beckman Coulter: anti-CD45-PC7 (J33), anti-CD3-ECD (UCHT1), anti-CD16-FITC (3G8), and anti-CD56-PE (N901). These were added to 30 μL of whole blood, and the mixture was incubated for 15 min at 4 °C. After adding 200 μL of optiLyse C Lysing solution (Beckman Coulter, Villepinte, France), the samples were analysed with a Beckman Coulter flow cytometer using FlowJo software (version 10.5.3, https://www.flowjo.com/solutions/flowjo/downloads, accessed on 10 November 2018). NK cells were further subdivided according to the intensity of the CD16 and CD56 expressions on the surfaces of NK cells.

### 2.3. Cytokine Profile Analysis

Serum samples were collected from all participants. A commercially available, human 14-plex assay kit was purchased from QuantoBio Technology, Beijing, China. The principle of the experiment is similar to that of a capture sandwich immunoassay: by coupling an antibody captured directly from the target protein to microspheres (4 μm or 5 μm) of 14 differing levels of APC fluorescence intensity, the target protein is bound to an individual microsphere and identified by a secondary biotinylated antibody. Streptavidin-PE (SA-PE, 20 μL) was added to each well and incubated for 30 min in the dark at 37 °C. This multiplex assay included cytokines involved in inflammation and angiogenesis pathways, such as IFN-γ, IL-1β, IL-2, IL-4, IL-5, IL-6, IL-8, IL-10, IL-12p70, IL-17A, IL-17F, IL-22, TNFα, and TNFβ. The mean fluorescence intensity of PE was detected using a Beckman Coulter flow cytometer.

### 2.4. Perforin and Granzyme B Content by Lymphocyte Cells

At 4 °C, 50 µL of whole blood was stained with a surface antibody cocktail (anti-CD45-PE-Cy7 (04A01), anti-CD3-PerCP (01A01), anti-CD8-APC-Cy7 (03A01), anti-CD16-APC (05A01), and anti-CD56-APC (06A01); all antibody volumes administered were 5 μL, Raisecare Biological Technology, Qingdao, China) for 20 min. After adding 200 μL of optiLyse C Lysing solution (Beckman Coulter, Villepinte, France), the samples were fixed and permeabilised with BD Cytofix/Cytoperm Buffer (BD Biosciences) for 15 min at room temperature (RT) in the dark, and they were washed with PBS twice. Finally, the cells were labelled with anti-perforin-FITC (δG9, BD Biosciences) and anti-granzyme B-PE antibodies (GB11, BD Biosciences) for 20 min at RT in the dark. After two washes, the cells were acquired using a Beckman Coulter and analysed using FlowJo version 10.5.3 software.

### 2.5. Lymphocyte Function

Phorbol 12-myristate 13-acetate (PMA)/ionomycin-stimulated lymphocyte function assays were performed as described previously [17]. A 100 µL volume of whole blood was diluted with 400 µL of an IMDM medium and incubated in the presence of Leukocyte Activation Cocktail with BD GolgiPlug (BD Biosciences, San Jose, CA, USA) for 4 h. Then, 300ul of cell supernatant was labelled with antibodies (anti-CD45-PerCP-Cy5.5 (2D1), anti-CD3-FITC (UCHT1), anti-CD4-APC-Cy7 (SK3), anti-CD8-PE (HIT8a), anti-CD16-PE-Cy7 (B73.1), and anti-CD56-PE-Cy7 (NCAM16.2), BD Biosciences) and incubated for 15 min in the dark at 37 °C. After adding 600 μL of optiLyse C Lysing solution (Beckman Coulter, Villepinte, France), the samples were then fixed and permeabilised. Lastly, the cells were labelled with an intracellular anti-IFN-γ-APC antibody (B27, BD Biosciences) and analysed using a BD FACSCanto II flow cytometer (BD Biosciences, San Jose, CA, USA). The percentage of IFN-γ+ cells in the different cell subsets defined their function.

### 2.6. Statistical Analyses

A statistical analysis was performed using SPSS version 22.0 and GraphPad Software 9.0. Continuous variables are expressed as mean ± SEM or interquartile range (IQR), depending on whether the data conformed to normal distribution. The statistical significance for comparisons between groups was determined using Student’s *t*-test or ANOVA. The Mann–Whitney U test (nonparametric) for independent samples was used to compare continuous variables. Differences between categorical variables were evaluated using contingency tables (χ^2^ test or Fisher’s exact test). All *p*-values were two-tailed, and differences with *p* < 0.05 were considered statistically significant.

## 3. Results

### 3.1. Baseline Characteristics of Patients with Omicron BA.2 Infection

Patients with a confirmed Omicron BA.2 infection (*n* = 537) were hospitalised and isolated for treatment in our hospital. In total, 508 patients were included in this study. The median age of the included patients was 45.00 years (17.75–67.00), 239 of whom (47.07%) were men. Most people infected with Omicron BA.2 develop only asymptomatic or mildly symptomatic. Cough (261 (51.37%)) and fever (69 (13.58%)) were the most common symptoms. Other symptoms included sore throat (49 (9.64%)), expectoration (46 (9.06%)), pulmonary infection (43 (8.46%)), and nasal congestion (41 (8.07%)), among other symptoms. Hypertension (80 (15.75%)), diabetes mellitus (20 (3.94%)), chronic liver or kidney disease (22 (4.33%)), and pulmonary disease (15 (2.95%)) were the most common underlying illnesses (Table 1).

Of the 508 patients, 35 (6.89%) suffered from low fever, 27 (5.31%) from moderate fever, and 7 (1.38%) from high fever. We observed a milder body temperature elevation in the vaccinated patients than in the unvaccinated controls (*p* < 0.001). On hospital admission, the symptoms of the patients without vaccination were significantly more severe than those of the patients who had received COVID-19 vaccines: shortness of breath, 4 (0.99%) vs. 5 (4.90%), *p* = 0.012; chest tightness, 14 (3.45%) vs. 9 (8.82%), *p* = 0.016; nausea or vomiting, 3 (0.74%) vs. 4 (3.92%), *p* = 0.031; abdominal pain, 1 (0.25%) vs. 3 (2.94%), *p* = 0.014; headache, 6 (1.48%) vs. 7 (6.86%), *p* = 0.003; and pulmonary infection, 26 (6.40%) vs. 17 (16.67%), *p* < 0.001, respectively. Compared to the vaccinated patients, the unvaccinated patients tended to report cough, nasal congestion, sore throat, fatigue, expectoration, shortness of breath, and chest pain more frequently. There were more patients with diabetes mellitus in the unvaccinated group than in the vaccinated group (7 (6.86%) vs. 13 (3.20%), *p* = 0.069). However, there were no significant differences between the two cohorts with regard to hypertension, coronary heart disease, allergies, and underlying pulmonary diseases (Table 1). The analyses showed that vaccination, as a sole intervention, can be effective in mitigating the impact of COVID-19 outbreaks.

### 3.2. Lymphocyte Subsets and Cytokines of Patients with Omicron BA.2 on Hospital Admission

For all patients infected with Omicron BA.2 virus, the lymphocyte subsets and cytokines in the peripheral blood samples were examined using flow cytometry. No statistical differences or trend changes were observed in the absolute cell numbers, the percentages of T- and B-lymphocyte subsets for both unvaccinated and vaccinated groups, or the CD4^+^/CD8^+^ ratio, but we observed an increase in circulating CD3^−^CD56^+^CD16^+^ NK lymphocytes (the absolute cell numbers of vaccinated vs. unvaccinated patients, 244.26 cells/μL [139.83–364.03] vs. 178.35 cells/μL [100.43–330.00], *p* = 0.001; the percentages of vaccinated vs. unvaccinated patients 17.51% [12.36–24.69] vs. 13.20 % [9.31–20.48], *p* < 0.000, respectively) (Figure 1A and Appendix A). IL-1β, IL-10, and TNF-α levels were significantly higher in the vaccinated group than in the unvaccinated group (4.39 pg/mL [3.33–8.74] vs. 3.50 pg/mL [2.98–4.20], *p* < 0.000; 4.66 pg/mL [3.90–8.18] vs. 4.08 pg/mL [3.55-4.95], *p* < 0.000; 5.30 pg/mL [4.18–9.00] vs. 4.61 pg/mL [3.86–5.86], *p* = 0.002, respectively) (Figure 1B and Appendix A). IFN-γ, IL-2, IL-4, IL-5, IL-6, IL-17A, IL-22, and TNF-β levels were higher in the vaccinated group than in the unvaccinated group, but they were still within the normal range. There was no significant difference between the two cohorts in terms of IL-8, IL-12p70, and IL-17F levels (Figure 1B and Appendix A). The results show that higher levels of NK cells and mild increases in both serum pro- and anti-inflammatory cytokine levels were observed in the patients infected with Omicron after vaccination.

### 3.3. Expansion of CD16brightCD56dim NK Cells Subsets during Omicron BA.2 Infection after Vaccination

NK cells are important components of the antiviral innate immune response, and different NK cell subsets play distinct roles [18]. We conducted an additional analysis of the NK cells in Omicron BA.2 infection, with or without vaccination, to further characterise the NK cell subsets and assess NK cell function using flow cytometry. Consistent with our results, on analysing the frequencies of the NK cells (CD56^+^ CD3^−^), we observed increased proportions of the CD56^bright^ subset of NK cells in the vaccinated group (Figure 2A, upper panel). A further assessment of the CD16^bright^CD56^dim^, CD16^dim^CD56^dim^, and CD16^+/−^CD56^bright^ NK cell subsets from a previous CD56^+^CD3^−^ gate (not showed) was undertaken. There was an increase in the CD16^bright^CD56^dim^ NK cell subsets with a decrease in the CD16^dim^CD56^dim^ NK cell composition in the vaccinated patients compared to the unvaccinated controls (Figure 2A, bottom panel). The right columnar graph summaries further confirm that all the trends discussed above were statistically significant (Figure 2B). Collectively, these results suggest dynamic changes in NK cell subsets in response to Omicron BA.2 infection after vaccination, with a significant expansion of CD16^bright^CD56^dim^ NK cells.

### 3.4. CD16brightCD56dim NK Cells Have Strong Potential in Cytokine Secretion and Cytotoxicity during Omicron BA.2 Infection after Vaccination

To determine the influence of Omicron BA.2 infection after COVID-19 vaccination on NK cell activity and cytotoxicity, we further evaluated the expressions of cytotoxic mediators, perforin and granzyme B. Additionally, the production of IFN-γ was also evaluated via intracellular cytokine staining after a short-term restimulation of NK cells in vitro using flow cytometry. The ability of NK cells to degranulate and release IFN-γ is a crucial immune defence against viral infections. As shown in Figure 3A, perforin, granzyme B, and IFN-γ levels were significantly higher in the vaccinated group than in the unvaccinated group. This is consistent with our results, indicating that the higher the percentage of CD16^bright^CD56^dim^ NK cells, the higher the production of perforin, granzyme B, and IFN-γ in the vaccinated group. The histogram shows a statistical graph (Figure 3B). Additionally, there was a trend towards significance in perforin, granzyme B, and IFN-γ levels in the CD8^+^ and CD4^+^ T cells in the vaccinated group, but this difference was not statistically significant (Appendix A). Altogether, these results indicate that CD16^bright^CD56^dim^ NK cells have potent cytokine secretion and cytotoxicity during Omicron BA.2 infection after vaccination.

## 4. Discussion

The coronavirus disease (COVID-19) pandemic continues to evolve along with its causative agent SARS-CoV-2. The Omicron variant BA.2 of SARS-CoV-2 has rapidly become the dominant variant worldwide [9]. Moreover, Omicron BA.2, which possesses an alarming number of mutations (>30), has raised concerns about the reduced effectiveness of vaccines and antibodies against these variants [19,20]. The Omicron variant BA.2 shows characteristics different from those of previous variants, and it is highly infectious, highly transmissible, minimally pathogenic, and vaccine- and antibody-tolerant [21]. A study has shown the significant immune escape of the Omicron variant in COVID-19 convalescent patients infected with the original SARS-CoV-2 strain [22]. Another study has shown that Omicron thwarts some of the world’s most-used COVID vaccines [23]. Different countries and medical institutions, as well as different subpopulations and age groups, may benefit from different vaccine products developed on various platforms. Different types of vaccines may stimulate different antibodies and immune responses against the Omicron variant BA.2 [13]. Inactivated vaccines have catered to almost half of the world’s COVID-19 vaccine needs [23], and they remain crucial for preventing hospitalisation and death from COVID-19.

While most people are actively vaccinated, many gaps remain in terms of our understanding of the immune reactivity to the Omicron variant BA.2. In the present study, all individuals infected with Omicron BA.2 were divided into two groups: an unvaccinated group and a vaccinated group. The two groups did not differ in age but had a wide age range. The control group mainly comprised the elderly and children under 3 years old who were not vaccinated. Patients aged under 3 years old do not meet the technical guidelines for COVID-19 vaccination. Elderly individuals with chronic underlying diseases may not benefit from vaccination, or they may refuse vaccination. Because this is a retrospective, single-centre study, the collected data are somewhat limited. Despite clinical symptoms being exhibited in both groups, vaccination led to a significant decline in nausea or vomiting, abdominal pain, headache, pulmonary infection, and overall clinical symptoms and a moderate rise in body temperature. This finding may be consistent with previous studies showing that an inactivation-based COVID-19 vaccine induces cross-neutralising immunity against the SARS-CoV-2 Omicron variant [24,25] and alleviates disease symptoms [26]. In addition, we found significantly higher levels of IL-1β, IL-10, and TNF-α and mildly elevated levels of IFN-γ, IL-2, IL-4, IL-5, IL-6, IL-17A, IL-22, and TNF-β in patients after vaccination, but they remained within the normal range. There was no significant difference between the two cohorts in terms of IL-8, IL-12p70, and IL-17F levels. It is now generally accepted that CD4+ T cells are functionally divided into various subsets: Th1 (IFN-γ, GM-CSF, IL-2, and TNF-β/α), Th2 (IL-4, IL-5, IL-9, IL-10, and IL-13), Th17 (IL-17A/F, IL-21, IL-22, IL-26, GM-CSF, and TNF-α), Th9 (IL-9 and IL-21), and Treg (IL-10 and TGF-β) [27]. Similarly, when characterising the Th1/Th2/Th17 cytokine profiles among the different stages of COVID-19 infection, it was found that most of them were elevated in patients with COVID-19 but were not statistically significant, except for pro-inflammatory IL-6 [28] or Th1 cytokines [29]. In addition, there was no significant difference in the basal secretion of IFN-γ, TNF-α, or IL-2 between the vaccinated subjects [30]. These studies reveal that SARS-CoV-2-specific T cells predominantly produced Th1 cytokines to promote cellular immunity. Although the mutation and evolution of COVID-19 accelerate transmission and infectivity, severe disease appears to be relatively uncommon in individuals infected with the Omicron variant BA.2 [21]. This is due to a combination of the internal and external factors of Omicron BA.2 virus. The extrinsic reason is that Omicron infections tend to be mild in people who have been previously infected, those who have been vaccinated, and the young, who are associated with the presence of antibodies already in the body and fewer underlying diseases [24,31]. The intrinsic reason is that Omicron variant BA.2 replicates less well in the lungs and binds more strongly to the mucosal epithelium of the nose, mainly infecting the upper respiratory tract and not the lower respiratory tract [32]. These analyses highlight that vaccination, as a sole intervention, can be effective in mitigating the impact of substantial SARS-CoV-2 outbreaks.

Many studies have shown that T-cell responses to the SARS-CoV-2 spike cross-recognise Omicron [33,34,35,36]. SARS-CoV-2 spike-specific CD4^+^ and CD8^+^ T cells induced by prior infection or COVID-19 vaccination provide extensive immune coverage against the Omicron [33,34,35,37] and Delta strains [30]. The majority of T-cell epitopes are unaffected by mutations in these variant strains [38]. This evidence points to the potential role of T cells in alleviating COVID-19 severity and contributing to disease protection. However, there were no significant differences or trend changes between the T- and B-lymphocyte subsets, while a significant expansion of NK lymphocytes in the COVID-19-vaccinated patients was observed. According to our above research results, we hypothesise that the vaccine group produces antibodies to mediate ADCC through the binding of the Fc segment of the antibody to the FcR of NK cells. Furthermore, there is other evidence that both cellular and humoral immune responses after vaccination are stronger than those after naturally occurring infection, pointing out the need for immune activity elicited by vaccines to overcome the Omicron pandemic [30]. Several reasons may lead to inconsistent conclusions from different studies.

NK cells are important effectors of innate immunity and play a critical role in antiviral infections [39]. To further examine changes in NK cell subsets and function, the effects of the cytokine secretion and cytotoxicity of NK cell subsets in defense against Omicron BA.2 viral infections after vaccination were evaluated. Our study revealed that the CD16^bright^CD56^dim^ NK cell subsets showed increased functional competence, as evidenced by a significantly greater IFN-γ production that was consistent with the above detection of cytokines in plasma and stronger cytotoxic capabilities in the individuals infected with Omicron BA.2 after vaccination. Alejandro et al. observed an expansion of the CD56^dim^CD16^neg^ NK subset and lower cytotoxic capacities in patients with COVID-19 [40]. Our results agree with reports stating that SARS-CoV-2-specific NK-cell-mediated antibody-dependent cell-mediated cytotoxicity (ADCC) responses were subjected to NK cell FcγRIIIa genetic variants [41] and that the normal activity of NK cells might improve the control of COVID-19 by fighting the virus and suppressing fibrosis progression [42,43]. Briefly, NK cells exhibit anti-SARS-CoV-2 activity. In contrast, the induction of NK-cell-mediated ADCC against SARS-CoV-2 after natural infection is more potent than that after vaccination [44]. Alejandro et al. observed an expansion of the CD56^dim^CD16^neg^ NK subset and lower cytotoxic capacities in patients with COVID-19 [40]. Although Omicron-based immunogens might be powerful enablers, they are unlikely to substantially outperform existing vaccines for priming SARS-CoV-2-naive individuals [45]. NK cells elicited by vaccination are cross-reactive with Omicron, most likely contributing to the maintenance of vaccine effectiveness against severe disease after Omicron infection.

In summary, the present study shows that the CD16^bright^CD56^dim^ NK cell subset was redistributed and activated to counteract the SARS-CoV-2 Omicron subvariant BA.2 after vaccination. Nevertheless, this study has some limitations. Elderly individuals with chronic underlying diseases that may aggravate immune dysfunction were included. Since this is a retrospective study, we cannot exclude some of the overlapping associations of chronic underlying diseases with immune dysfunction. Future multicentre prospective studies with adequate sample sizes will be required to validate the obtained data. We did not perform a more detailed categorisation based on the dose and time of vaccination, the different types of vaccines, or the days of hospitalisation. Moreover, further investigation is needed to gain insights into the impact of CD16^bright^CD56^dim^ NK-cell-mediated ADCC on the immune pathogenesis of the Omicron subvariant BA.2. Overall, a better understanding of vaccine-induced CD16^bright^CD56^dim^ NK cell subsets may offer some insights into therapeutic strategies for the treatment of Omicron BA.2 infections.

## Figures and Tables

**Figure 1 microorganisms-11-00940-f001:**
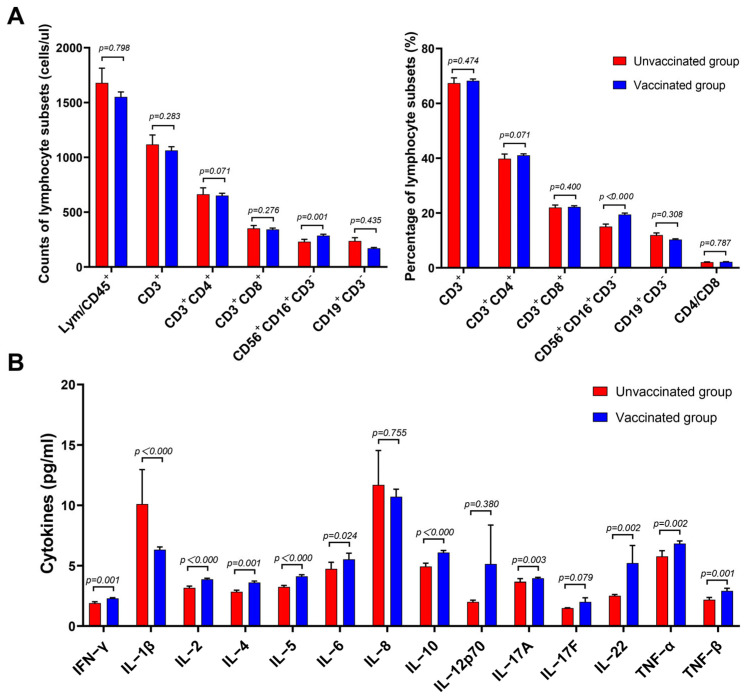
Comparison of the lymphocyte subsets and cytokines between the unvaccinated and vaccinated groups. (**A**) Absolute numbers and percentage of lymphocyte subsets and the CD4^+^/CD8^+^ T-cell ratio. (**B**) Graphical representation of the concentrations of different cytokines in the unvaccinated and vaccinated groups. Unvaccinated group, *n* = 102; vaccinated group, *n* = 406. All *p* values were two-tailed, and differences with *p* < 0.05 were considered statistically significant. ns, not significant.

**Figure 2 microorganisms-11-00940-f002:**
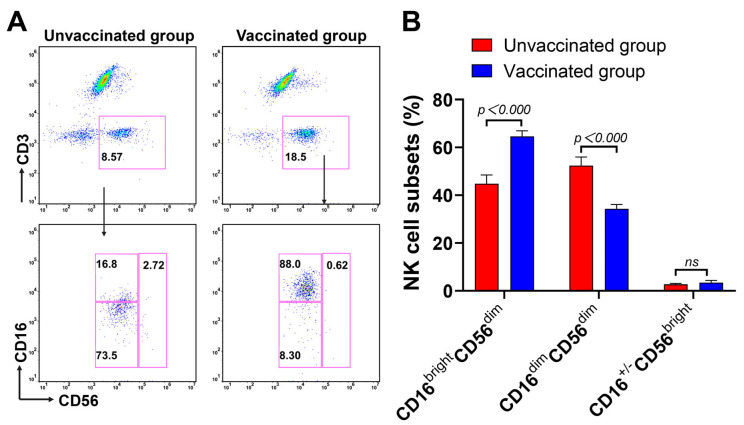
Comparison of the NK cell subsets between the unvaccinated and vaccinated groups. (**A**) The expressions of CD16 and CD56 on NK cells were measured using flow cytometry. (**B**) Bar graph of the proportions of the NK cell subsets. Unvaccinated group, *n* = 96; vaccinated group, *n* = 347. All *p*-values were two-tailed, and differences with *p* < 0.05 were considered statistically significant. *ns*, not significant.

**Figure 3 microorganisms-11-00940-f003:**
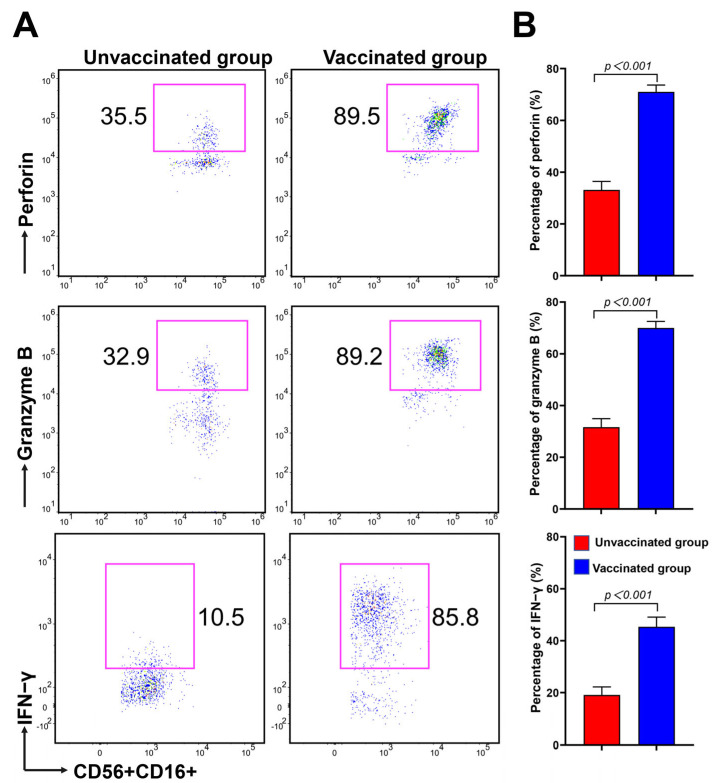
Comparison of perforin, granzyme B, and IFN-γ on the NK cell subsets between the unvaccinated and vaccinated groups. (**A**) The expressions of perforin, granzyme B, and IFN-γ on CD56^+^CD16^+^ NK cells were evaluated using flow cytometry. (**B**) Graphical representation of perforin, granzyme B (unvaccinated group, *n* = 18; vaccinated group, *n* = 50), and IFN-γ (unvaccinated group, *n* = 11; vaccinated group, *n* = 45) expressions on CD56^+^CD16^+^ NK cells. All *p*-values were two-tailed, and differences with *p* < 0.05 were considered statistically significant. *ns*, not significant.

**Table 1 microorganisms-11-00940-t001:** Comparison of the clinical characteristics between the unvaccinated and vaccinated groups.

Characteristic	Total (*n* = 508)	Unvaccinated Group (*n* = 102)	Vaccinated Group (*n* = 406)	** *p* ** **-Value**
Age, median (IQR *), years	45.00 (17.75–67.00)	33.00 (3.50–70.50)	47.00 (19.00–66.00)	0.066
Male/female, *n* (%)	239 (47.04)/269 (52.95)	49 (48.04)/53 (51.96)	190 (46.80)/216 (53.20)	0.822
Fever				<0.001
Low (37.3–38 °C), *n* (%)	35 (6.89)	12 (11.76)	23 (5.67)	
Moderate (38.1–39 °C), *n* (%)	27 (5.31)	16 (15.68)	11 (2.71)	
High (39–41 °C), *n* (%)	7 (1.38)	4 (3.92)	3 (0.74)	
Cough, *n* (%)	261 (51.37)	65 (63.73)	196 (48.28)	0.110
Expectoration, *n* (%)	46 (9.06)	11 (10.78)	35 (8.62)	0.696
Nasal congestion, *n* (%)	41 (8.07)	11 (10.78)	30 (7.39)	0.202
Chills, *n* (%)	3 (0.59)	1 (0.98)	2 (0.49)	1.000
Sore throat, *n* (%)	49 (9.64)	14 (13.72)	35 (8.62)	0.139
Swollen tonsils, *n* (%)	4 (0.78)	1 (0.98)	3 (0.74)	1.000
Myalgia, *n* (%)	13 (2.56)	3 (2.94)	10 (2.46)	1.000
Fatigue, *n* (%)	9 (1.77)	2 (1.96)	7 (1.72)	1.000
Anorexia, *n* (%)	3 (0.60)	1 (0.98)	2 (0.50)	1.000
Short of breath, *n* (%)	9 (1.77)	5 (4.90)	4 (0.99)	0.012
Chest tightness, *n* (%)	23 (4.53)	9 (8.82)	14 (3.45)	0.016
Chest pain, *n* (%)	3 (0.60)	1 (0.98)	2 (0.49)	1.000
Nausea or vomiting, *n* (%)	7 (1.38)	4 (3.92)	3 (0.74)	0.031
Abdominal pain, *n* (%)	4 (0.79)	3 (2.94)	1 (0.25)	0.014
Diarrhoea, *n* (%)	2 (0.40)	1 (0.98)	1 (0.25)	0.806
Headache, *n* (%)	13 (2.56)	7 (6.86)	6 (1.48)	0.003
Dizziness, *n* (%)	28 (5.51)	4 (3.92)	24 (5.91)	0.586
Pulmonary infection, *n* (%)	43 (8.46)	17 (16.67)	26 (6.40)	<0.001
Hypertension, *n* (%)	80 (15.75)	20 (19.61)	60 (14.78)	0.252
Coronary heart disease, *n* (%)	14 (2.76)	5 (4.90)	9 (2.21)	0.139
Cerebral peduncle, *n* (%)	14 (2.76)	4 (3.92)	10 (2.46)	0.554
Diabetes mellitus, *n* (%)	20 (3.94)	7 (6.86)	13 (3.20)	0.069
Allergy, *n* (%)	14 (2.76)	4 (3.92)	10 (2.46)	0.598
Underlying pulmonary diseases, *n* (%)	15 (2.95)	5 (4.90)	10 (2.46)	0.147
Chronic liver or kidney disease, *n* (%)	22 (4.33)	7 (6.86)	15 (3.69)	0.136
Malignancy, *n* (%)	10 (1.97)	4 (3.92)	6 (1.48)	0.182

* IQR: interquartile range.

## Data Availability

The data presented in this study are available on request from the corresponding author. The data are not publicly available in accordance with the guidelines of the Ethics Committee.

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
