# Peer review of "Redistribution and Activation of CD16brightCD56dim NK Cell Subset to Fight against Omicron Subvariant BA.2 after COVID-19 Vaccination"

_microorganisms, 2023, doi:10.3390/microorganisms11040940_

Round 1
Reviewer 1 Report
The authors presented an interesting a relevant evaluation of CoV19 vaccine efficiency against BA2 infection. The questions asked are pertinent, as is they mechanistic insight regarding cellular involvement in protection. There study population was well designed and appropriate, though the investigators should better describe the type of vaccines used in the patient population. Also, analyses could have placed more granular focus onto male versus female responses to infection/protection from signs of disease.
Author Response
Point 1: The authors presented an interesting a relevant evaluation of CoV19 vaccine efficiency against BA2 infection. The questions asked are pertinent, as is they mechanistic insight regarding cellular involvement in protection. There study population was well designed and appropriate, though the investigators should better describe the type of vaccines used in the patient population. Also, analyses could have placed more granular focus onto male versus female responses to infection/protection from signs of disease.
Response 1: It is our great honor to have your affirmation. Unfortunately, since this is a retrospective study, further material from these patients is no longer available due to lockdown. Therefore, to fully explore the issue that male versus female responses to infection/protection from signs of disease will require a future study, as this is ‘next-step’ question that will require significant new experimental effort. According to Quality of English Language from you, we polished the manuscript with a professional assistance in writing, conscientiously.

Reviewer 2 Report
1. The authors described that the infected cases were BA.2 which was detected by nucleic acid testing. Which types of nucleic acid testing did the authors used? Did your system confirm that all cases were BA.2. Only sequencing can definitely identify the variants. How did your test confirm that all cases were BA.2 Please describe clearly to be sured that all the cases were BA.2.
2. Regarding the COVID-19 vaccination, the authors did not describe that what types of COVID-19 vaccines were injected? The authors described up to three doses of vaccines were received. Is it heterologous or homologus vaccination Please describe it.
3.In this study, 102 patients were unvaccinated. Why did the patients did not receive vaccine? Is it ethical to use control although vaccines are available and the number of unvaccinated were so high. Please make discussion for it.
4. Regarding the ethical approval, it received in 2023. The patients were from 2022. Is it retrospective study or the authors used the left-over samples Please describe it. If the retrospective study, how did the authors take informed consent?
5. The type of vaccines are very important for B cell immunity and T cell immunity for the vaccinated people. The authors must clearly describe the types of vaccine injected in the study participants and should analyse based on the type of vaccination. Furthermore, the authors need to describe the vaccines were omicron variant specific or not?
6. The authors described that there are more patients with diabetes mellitus in the control group. DM is a risk for severe diseases and why those patients were also immune dysfunction? Your study was the study on immune status. It will be confounding factor for this study? Please discuss it.
7. The authors described that all the patients were mild clinical diseases. Why did the authors did not study for severe cases? Please discuss it.
Author Response
Point 1: The authors described that the infected cases were BA.2 which was detected by nucleic acid testing. Which types of nucleic acid testing did the authors used? Did your system confirm that all cases were BA.2. Only sequencing can definitely identify the variants. How did your test confirm that all cases were BA.2 Please describe clearly to be sured that all the cases were BA.2.
Response 1: Thank you for your comments. The Oral-nasopharyngeal swabs was tested using the COVID-19 (ORF lab/N gene) nucleic-acid detection kit (Da An Gene Co., Ltd., Guangzhou, China and Bio-Germ Co., Ltd., Shanghai, China). All cases were sent to the Jiangxi provincial CDC for confirmation testing. The Illumina Nextseq 550 sequencing platform was used for further sequencing.
Correspondingly, we add the description in the related content in line 87 and 89-90.
Point 2: Regarding the COVID-19 vaccination, the authors did not describe that what types of COVID-19 vaccines were injected? The authors described up to three doses of vaccines were received. Is it heterologous or homologus vaccination Please describe it.
Response 2: Thank you for your comment. Whole, inactivated, vaccine-homologous COVID-19 viruses were administered at three time points from the first dose to the third dose. As you suggested, we have added this in line 94-95.
Point 3: In this study, 102 patients were unvaccinated. Why did the patients did not receive vaccine? Is it ethical to use control although vaccines are available and the number of unvaccinated were so high. Please make discussion for it.
Response3: Thank you for giving us a chance to explain it. It is mainly the elderly and under 3 years old children who are not vaccinated. While patients aged under 3 years old don’t meet the technical guidelines for COVID-19 vaccination. Elderly individuals with chronic underlying diseases would not benefit from vaccination or their refusal of vaccination. Remaining barriers to vaccination included only a tiny minority of adults perception that healthy adults were not vaccinated due to vaccine inaccessibility, not trusting the nation recommendation for vaccination, and not knowing where to get vaccinated. Also, we have modified the text as you suggested (line 306-311 in the text).
Point 4: Regarding the ethical approval, it received in 2023. The patients were from 2022. Is it retrospective study or the authors used the left-over samples Please describe it. If the retrospective study, how did the authors take informed consent?
Response 4: Thank you for your comment. The First Affiliated Hospital of Nanchang University, Jiangxi, China, is a large, integrated, research-based tertiary hospital integrating medical treatment, teaching, and research. In our hospital administration system, we have added informed consent statements for every medical treatments and examinations, which are maintained in the an electronic medical record system (EMRS). This is a retrospective study to collect patient’s clinical examination data in 2022. It takes some time for the Ethics Committee of the First Affiliated Hospital of Nanchang University to review and approve this study. Approval Code: (2023) CDYFYYLK (01-019); Approval Date: January 29th, 2023
Point 5: The type of vaccines are very important for B cell immunity and T cell immunity for the vaccinated people. The authors must clearly describe the types of vaccine injected in the study participants and should analyse based on the type of vaccination. Furthermore, the authors need to describe the vaccines were omicron variant specific or not?
Response 5: Special thanks to you for your good comments. I have answered this question in Point 2 above. Whole, inactivated, vaccine-homologous COVID-19 viruses were administered at three time points from the first dose to the third dose. However, the vaccine are not specific for omicron variant. Correspondingly, we add the description in the related content in line 314-315.
Point 6: The authors described that there are more patients with diabetes mellitus in the control group. DM is a risk for severe diseases and why those patients were also immune dysfunction? Your study was the study on immune status. It will be confounding factor for this study? Please discuss it.
Response 6: Thank you for your comment. As I have explained above, it is mainly the elderly and under 3 years old children who are not vaccinated in the control group. Elderly individuals with chronic underlying diseases such as hypertension, coronary heart disease, cerebral peduncle, diabetes and so on, these may aggravate immune dysfunction. Since this is a retrospective study, not designed to study co-morbidities of MD and COVID-19 diseases, we cannot exclude that some of the overlapping association of MD with immune dysfunction. Future multicenter prospective studies with adequate sample sizes will be required to validate the data obtained.This has been addressed by the following line 379-383.
Point 7: The authors described that all the patients were mild clinical diseases. Why did the authors did not study for severe cases? Please discuss it.
Response7: Thank you very much to point out the issues in our manuscript. Actually, no serious cases have occurred in our hospital. Omicron variant BA.2 shows dierent characteristics from previous variants, which is highly infectious, highly transmissible, minimally pathogenic, vaccine and antibody tolerant; however, it is less likely to cause severe illness, resulting in fewer deaths. The cause is a combination of internal and external causes of the virus. The extrinsic reason is that Omicron infections tend to be mild in people who have been previously infected, those who have been vaccinated, and the young (more of whom are in this wave), which is associated with the presence of antibodies already in the body and fewer underlying diseases. The intrinsic reason is that Omicron variant BA.2 replicates less well in the lungs and binds more strongly to the mucosal epithelium of the nose, which it mainly infects the upper respiratory tract, not the lower respiratory tract. Therefore, effective defense may be the key to controlling the epidemic today, rather than just “living with” the virus. Also, we have modified the text as you suggested (line 333-339 in the text).
